# Ensemble optimal interpolation for adjoint-free biogeochemical data assimilation

**Jann Paul Mattern***, **Christopher A. Edwards**

Ocean Sciences Department, UC Santa Cruz, Santa Cruz, CA, United States of America

* jmattern@ucsc.edu

## Abstract

Advanced marine ecosystem models can contain more than 100 biogeochemical variables, making data assimilation for these models a challenging prospect. Traditional variational data assimilation techniques like 4dVar rely on tangent linear and adjoint code, which can be difficult to create for complex ecosystem models with more than a few dozen variables. More recent hybrid ensemble-variational data assimilation techniques use ensembles of model forecasts to produce model statistics and can thus avoid the need for tangent linear or adjoint code. We present a new implementation of a four-dimensional ensemble optimal interpolation (4dEnOI) technique for use with coupled physical-ecosystem models. Our 4dEnOI implementation uses a small ensemble, and spatial and variable covariance localization to create reliable flow-dependent statistics. The technique is easy to implement, requires no tangent linear or adjoint code, and is computationally suitable for advanced ecosystem models. We test the 4dEnOI implementation in comparison to a 4dVar technique for a simple marine ecosystem model with 4 biogeochemical variables, coupled to a physical circulation model for the California Current System. In these tests, our 4dEnOI reference implementation performs similarly well to the 4dVar benchmark in lowering the model observation misfit. We show that the 4dEnOI results depend heavily on covariance localization generally, and benefit from variable localization in particular, when it is applied to reduce the coupling strength between the physical and biogeochemical model and the biogeochemical variables. The 4dEnOI results can be further improved by small modifications to the algorithm, such as multiple 4dEnOI iterations, albeit at additional computational cost.

## 1 Introduction

Data assimilation (DA) systems rigorously constrain numerical models using observed data, aiming to produce the most accurate estimate of the modeled state. Among many applications to dynamical models in the geosciences, DA is applied to coupled physical-biogeochemical ocean models, where it yields large improvements in the model's state estimates (see, e.g., [1–8]). Marine physical-biogeochemical models can be a difficult target for DA, because they are high dimensional, due to numerous state variables, undergo frequent modifications and expansion by modelers, and are characterized by a high level of nonlinearity. Variational DA techniques, one group of widely-used DA techniques (see, e.g., [5, 7, 9] for applications to coupled physical-biogeochemical models), require tangent-linear and adjoint code which are

**Data Availability Statement:** All data used in the analysis are available from public data sources. Please see Table 1 for the relevant source details.

**Funding:** JPM and CAE were both funded by the Simons Foundation (CBIOMES award ID: 549949).

https://cbiomes.org/ The funders had no role in study design, data collection and analysis, decision to publish, or preparation of the manuscript.

**Competing interests:** The authors have declared that no competing interests exist.

based on the derivative of the regular, nonlinear model code and are labor-intensive to obtain for complex models and to maintain when the models undergo frequent changes. Ensemble-based DA techniques, the second group of widely used DA techniques (see, e.g., [1, 10, 11] for applications to coupled physical-biogeochemical models), rely on statistics derived from ensembles of model simulations to improve model state estimates. While ensemble-based techniques are typically easier to implement, they can have large computational costs associated with creating large ensembles and computing ensemble statistics for a high-dimensional model state. Hybrid techniques, that combine elements of variational and ensemble-based DA, exist and are becoming more prevalent [12, 13]. Based on the technique, they share advantages and disadvantages of the underlying methodologies they are based on.

With the goal of applying joint physical-biogeochemical DA to a current biogeochemical model with more than 30 state variables, we searched for a DA technique with two characteristics: (1) relatively easy to implement for a complex biogeochemical model undergoing frequent code changes, and (2) reasonable computational requirements in terms of processing and memory to make it practical for current high-performance computers. The 4-dimensional variational DA technique (4dVar) yields large improvements in the biogeochemical model state estimates [5], and it fulfills our computational requirements (a factor of $\approx 200$ increase in run-time compared to a simulation without DA for our biogeochemical model implementation in [5]), yet it is very time-consuming to implement or run for complex biogeochemical models. An implementation requires either hand-coding the required tangent linear and adjoint code, which is cumbersome and error-prone, or reliance on automatic differentiation (e.g., [14]). Automatic differentiation that is based on the successive differentiation of each operation in the biogeochemical model code requires specialized software and is not straightforward to implement. An alternative is to use the mathematical construct of dual numbers for automatic differentiation. Dual numbers can be used to automatically obtain the exact value of a function derivative without explicitly computing the derivative, and can be implemented with comparatively little effort [15], but the computational cost becomes prohibitive as the number of biogeochemical variables grows.

Ensemble-based DA techniques, with the Ensemble Kalman Filter (EnKF) the best known among them, have no requirements for tangent-linear or adjoint code but can be computationally demanding. One of the most computationally efficient ensemble-based techniques is the Ensemble Optimal Interpolation technique (EnOI, [16–18]). What makes EnOI additionally attractive for biogeochemical data assimilation is that it is easy to implement and is easily able to accommodate changes to the biogeochemical code. The EnOI implementation originally introduced in [19] used a static ensemble to update a single model state. It is presented as a computationally less demanding alternative to the EnKF, where a dynamical ensemble is used to update the full ensemble.

With the purpose of assimilating physical and biogeochemical observations jointly into a coupled physical-biogeochemical model, we implement a version of EnOI, that retains one of the main characteristics of the EnOI presented in [19]: using a static ensemble, only a single model state is updated by the DA. Unlike most previous approaches, we use EnOI for asynchronous, or 4-dimensional DA, updating the model with observations that are distributed in time across a given window; for this reason, we refer to our implementation as "4dEnOI". What sets our implementation further apart from other 4dEnOI approaches, such as [20], is that the first ensemble member is included in the computation of the ensemble-based statistics, which are otherwise only based on the static ensemble of model simulations. Even though a mostly static ensemble is used, the DA increment is based on flow-dependent statistics, where spatial and variable localization are used to obtain more reliable statistics. Here, spatial localization refers to the practice of dampening ensemble-based estimates of correlations if they are

spatially distant, which is commonly employed in ensemble-based data assimilation applications to correct the effects of small ensemble sizes [1, 21, 22]. Variable localization performs a similar function, but reduces the correlations between "dynamically distant" model variables [23]. Our 4dEnOI implementation is computationally efficient, allowing for division of the state space into small segments that can be processed individually. This subdivision reduces memory requirements and offers straightforward parallelization options (see S1 File for details). Its computational cost grows linearly with the number of variables, making it suitable for complex biogeochemical models with many variables and consequently a very high-dimensional state vector.

Though our 4dEnOI approach is ultimately intended to support a complex biogeochemical model, we test its implementation here using a relatively simple coupled physical-biogeochemical model that includes a Nutrient-Phytoplankton-Zooplankton-Detritus (NPZD) model with 4 biogeochemical variables. For this model, we have existing tangent linear and adjoint code [5], allowing us to compare the EnOI implementation to a 4dVar-based DA system. In the following, we describe the most important aspects of our 4dEnOI implementation (Section 2.1), emphasizing the way the dynamical ensemble is generated (Section 2.2) and the way spatial localization (Section 2.3) and variable localization (Section 2.4) are implemented.

## 2 Methods

### 2.1 4dEnOI implementation

We implemented an EnOI technique that is 4-dimensional and uses flow-dependent statistics. That is, our 4dEnOI implementation performs an update of the model state based on observations that vary both in space and time (and are thus 4-dimensional), and model dynamics (flow) are taken into account when computing the ensemble-based statistics that are needed to perform the update (Fig 1). The implementation is based on the `ROMSEnsemble.jl` package [24], written in the Julia language [25].

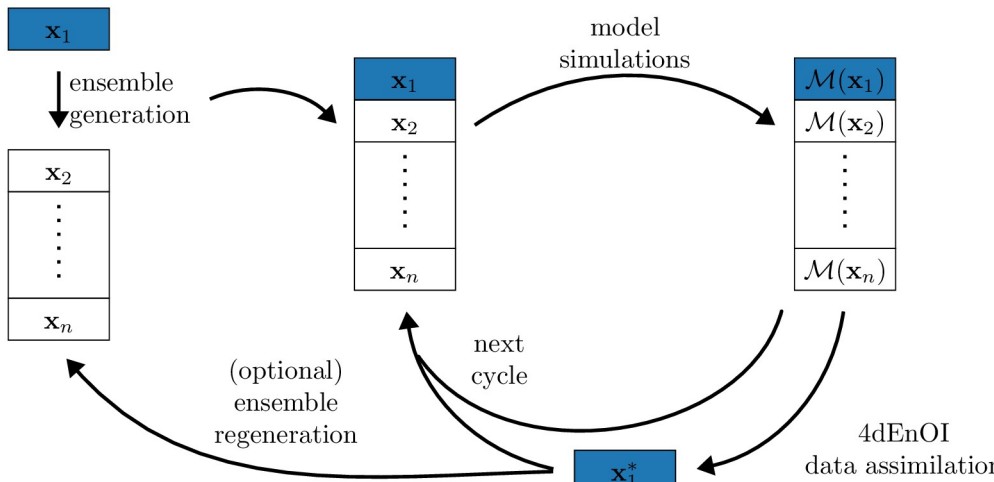

**Fig 1. Schematic of the EnOI implementation.** The first ensemble member $\mathbf{x}_1$ is used to generate the initial ensemble and regenerate it when desired (see Section 2.2). If the ensemble is not regenerated, a static ensemble for $\mathbf{x}_2$ to $\mathbf{x}_{n_{ens}}$ can be used. The coupled physical-biogeochemical model is used to run the ensemble forward to the next cycle, $\mathcal{M}$ denotes the full model trajectory for the given cycle. In each cycle, the 4dEnOI data assimilation uses statistics derived from the full ensemble to update $\mathbf{x}_1$, producing $\mathbf{x}_1^*$. The remaining ensemble members are run forward but not updated using DA.

For 4dEnOI, the DA update used to increment the prior model state $\mathbf{x}$ with observations $\mathbf{y}$ to obtain the posterior state $\mathbf{x}^*$, can be written as

$$\mathbf{x}^* = \mathbf{x} + (\alpha\,\mathbf{L} \circ \mathrm{cov}(\mathbf{X}, H\mathbf{X}))(\alpha\,\mathbf{L}' \circ \mathrm{cov}(H\mathbf{X}, H\mathbf{X})) + \mathbf{R})^{-1}(\mathbf{y} - H\mathbf{x}), \tag{1}$$

where $\mathbf{X} \in \mathbb{R}^{n_{\text{state}} \times n_{\text{ens}}}$ is the ensemble of model states arranged in a matrix, and the observation operator $H$ projects one, or an ensemble of model states into observational space using the model, thereby providing a nonlinear mapping between $\mathbf{x}$ and $\mathbf{y}$. Thus, $H\mathbf{X} \in \mathbb{R}^{n_{\text{obs}} \times n_{\text{ens}}}$ are the ensemble of model results at the spatial and temporal observation locations, arranged in a matrix. When using a static ensemble, this approach requires storing the full model state (at the start of each DA cycle), and the model solutions at the observation locations for each static ensemble member. The function cov is the sample covariance function, providing ensemble-based estimates of the covariance of the prior model state. Matrices $\mathbf{L} \in \mathbb{R}^{n_{\text{state}} \times n_{\text{obs}}}$ and $\mathbf{L}' \in \mathbb{R}^{n_{\text{obs}} \times n_{\text{obs}}}$ are used for localization, and the symbol $\circ$ denotes the element-wise vector product. Localization reduces the effect of spurious correlations that may appear in the covariance terms in Eq (1) due to small ensemble sizes (see Section 2.3 below for implementation details). Finally, $\alpha \in\, ]0, 1]$ is a scaling factor that may be used to reduce the 4dEnOI increment diminishing a deleterious reduction in the ensemble spread, referred to as "ensemble collapse" or "filter inbreeding" [26], which has also been utilized in previous EnOI implementations [16, 27].

If, unlike in this study, non time-evolving model covariances are used, or if $H$ is linear or linearized, Eq (1) is commonly [16, 28] expressed as

$$\mathbf{x}^* = \mathbf{x} + (\alpha\,\mathbf{L} \circ \mathbf{B}\mathbf{H}^{\mathrm{T}})(\alpha\,\mathbf{L}' \circ \mathbf{H}\mathbf{B}\mathbf{H}^{\mathrm{T}} + \mathbf{R})^{-1}(\mathbf{y} - \mathbf{H}\mathbf{x}), \tag{2}$$

where $\mathbf{B}$ is the ensemble-based covariance matrix of the model state, usually referred to as the background error covariance matrix, and $\mathbf{H}$ is the linearized version of the observation operator $H$. In cases when the values of a non time-evolving, static $\mathbf{B}$ are not representative of the true model uncertainty, $\alpha$ is often reduced to a value below 1, decreasing the magnitude of an imperfect update. In our implementation, which uses a time-evolving covariance terms, we typically do not reduce the increment and set $\alpha = 1$.

Our 4dEnOI implementation uses time-evolving, flow-dependent statistics based on an ensemble of $n_{\text{ens}} = 25$ ensemble members. The ensemble, which is providing estimates of the true model covariances, is used to update the state estimate $\mathbf{x}$ with observations based on Eq (1). Unlike many other ensemble-based DA techniques like the EnKF, the remaining ensemble is not updated with data directly, and can remain static. In most of our experiments, we use a hybrid ensemble, consisting of 24 static ensemble members, augmented by a single dynamically adjusted ensemble member ($\mathbf{x}$ in Eq (1) which we from here on denote as $\mathbf{x}_1$, the first ensemble member of an otherwise static ensemble). The use of a hybrid ensemble is analogous to similar approaches for the EnKF (see, e.g., [29]), but relies only on a single non-static ensemble member which is required by the 4dEnOI in any case, and thus does not require additional model simulations. However, the use of a hybrid ensemble may increase the computational cost of the implementation, as it prohibits calculating the ensemble statistic in advance. The generation of the static ensemble is described in detail in Section 2.2. There, we further examine the option of not using a static ensemble and, instead, regenerating the ensemble after a number of DA cycles, bringing the ensemble closer to the updated model state and passively updating the full ensemble with observations.

## 2.2 Ensemble generation

Our implementation relies on a principal component analysis (PCA) to both initially generate the static ensemble, and to (optionally) regenerate an updated, non-static ensemble after a

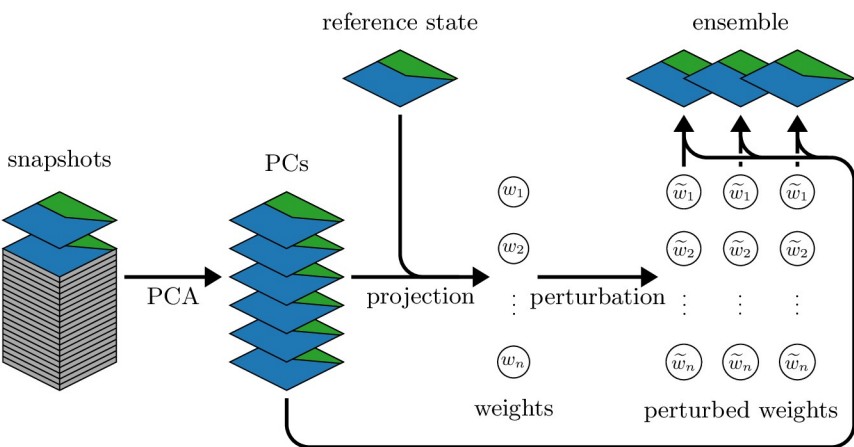

**Fig 2. Schematic of the ensemble generation.** Model snapshots are obtained from daily snapshots of a 5-year-long, non-assimilative model simulation; the initial model state (ensemble generation) or a successive model state (ensemble regeneration) is used as the reference solution from which the ensemble is created. In our implementation, we generate $n_{pc} = 25$ principal components (PCs) and an ensemble of $n_{ens} = 25$ members, including the reference solution.

number of DA cycles. To generate the ensemble, we use daily snapshots from a 5-year model simulation without DA to create a set of $n_{pc}$ leading principal components, a similar procedure to that presented in [20, 30, 31]. In brief, the initial model state (or the updated first ensemble member in case of a regeneration) is then projected onto the principal components, calculating a weight for each principal component. By adding noise to each weight and then multiplying each weight with the associated principal component, an ensemble of new model states is generated (Fig 2).

In more detail, to generate the ensemble, we first project the initial model state (or the first ensemble member, in case of a regeneration) $\mathbf{x}_1$ onto the first $n_{pc}$ leading principal components:

$$\mathbf{w} = \mathbf{V}^{\mathrm{T}} \, \mathbf{x}_1, \qquad (3)$$

where the matrix $\mathbf{V} \in \mathbb{R}^{n_{state} \times n_{pc}}$ contains the leading principal components in its columns. In the second step, the resulting vector of weights $w$ is perturbed with pseudo-random noise and then matrix-multiplied with $\mathbf{V}$ to obtain the remaining ensemble

$$\mathbf{x}_i = \mathbf{V} \, (\mathbf{w} \circ \mathbf{r}_i) \ \text{for} \ i = 2, 3, \ldots, n_{ens}. \qquad (4)$$

Here, each $\mathbf{r}_i$ is a vector of pseudo-random noise. In our implementation, we use $n_{pc} = 25$ (accounting for $> 91\%$ of the variance) and a uniform distribution on [0.5, 1.5] for each element of $\mathbf{r}_i$. These parameters are application-specific and can be adjusted based on the size of the state vector $n_{state}$ and the available computing resources. In order to avoid negative values in any variables representing concentrations, we further enforce a minimum threshold of 0.001, setting all entries associated with the biogeochemical variables in $\mathbf{x}_i$ that are below this threshold to the threshold value.

An ensemble regeneration can be beneficial if the first ensemble member, the only ensemble member updated by observations (Fig 1), is steered away from the remaining, static ensemble. Hence, using the procedure described above to regenerate the ensemble around the first ensemble member after it has been updated, may provide more reliable statistics in the following DA cycles. Although the process of ensemble regeneration implies that the ensemble is no

longer static, and hence requires additional computational resources, we examine the benefits of ensemble regeneration in this application in Section 3.5.

## 2.3 Spatial localization

For models in which the size of the state space (its dimension) is much larger than the size of the ensemble, which is true here and for many model applications in the geosciences, localization has become an important tool in ensemble-based DA. By down-weighting entries in the covariance matrix that are associated with distant grid points, spatial localization reduces the impact of two effects that may deteriorate DA results. First, it reduces the effect of spurious correlations, and second, localization increases the rank of the ensemble with respect to the model [32, 33]. Spurious correlations arise from small ensemble sizes which can create unrealistic correlations, for example between spatially distant grid points, an effect that would diminish if the correlation was estimated with a larger sample (i.e., a bigger ensemble). Increasing the rank of the ensemble improves the condition number of the matrix that is inverted in Eq (1) (here the condition number acts as an indicator for the ensemble rank, in an actual implementation of Eq (1), a Cholesky decomposition of the matrix can significantly speed up the computation).

We implement localization using $\mathbf{L} \in [0, 1]^{n_{\text{state}} \times n_{\text{obs}}}$ and $\mathbf{L}' \in [0, 1]^{n_{\text{obs}} \times n_{\text{obs}}}$, two matrices of weights with values between 0 and 1, indicating loss of correlation due to spatial distance between each observation and each grid point in state space. That is, for $\mathbf{L}$, we compute

$$\mathbf{L} = \mathbf{L}_{\text{x}} \circ \mathbf{L}_{\text{y}} \circ \mathbf{L}_{\text{z}} \in [0, 1]^{n_{\text{state}} \times n_{\text{obs}}}, \tag{5}$$

with each $\mathbf{L}_{\text{x}}$, $\mathbf{L}_{\text{y}}$ and $\mathbf{L}_{\text{z}}$ containing the weights attributed to the distance in x, y and z-direction, respectively. In the x-direction,

$$\mathbf{L}_{\text{x}} = \exp\left(-\left(\mathbf{x}_{\text{obs}} \mathbf{1}_{n_{\text{state}}}^{\text{T}} - \mathbf{1}_{n_{\text{obs}}} \mathbf{x}_{\text{state}}^{\text{T}}\right)^2 / s_{\text{x}}^2\right), \tag{6}$$

where $\mathbf{x}_{\text{obs}}$ and $\mathbf{x}_{\text{state}}$ are the vectors containing the x-coordinates of observations and model grid points respectively and $\mathbf{1}_n$ is a vector of ones of length $n$. The parameter $s_{\text{x}}$ is a factor determining the length scale of the localization in x-direction. The localization in y- and z-directions, governed by $\mathbf{L}_{\text{y}}$ and $\mathbf{L}_{\text{z}}$, is performed analogously to the x-direction in Eq (6) using the length scale parameters $s_{\text{y}}$ and $s_{\text{z}}$.

In our implementation, we use a single horizontal length scale $s_{\text{x}} = s_{\text{y}} = 1°$, applied in both x- and y-direction (in our configuration, 1° is equivalent to 10 horizontal grid cells; we note that this choice is anisotropic horizontally, but produced satisfactory results), and a vertical length scale of $s_{\text{z}} = 300$ m. In order to select suitable length scale values, we employed a "half an order of magnitude approach": first, using no vertical localization, we tested horizontal length scales of 0.1°, 0.3°, 1°, and 3°, increasing by half an order of magnitude. We then selected the one that minimized the prior error in 8 DA cycles (run with a static ensemble in the same setup presented in Section 2.5). If the lowest or highest value had the lowest error, the series would have been extended. In the second step, using the 1° horizontal length scale, we selected a vertical length scale using the same procedure, testing values of 10 m, 30 m, 100 m, 300 m, and 1000 m.

The second localization matrix $\mathbf{L}'$ is computed analogously to $\mathbf{L}$ in Eq (5), considering only the spatial distance between the observation locations. The same horizontal and vertical lengths scales used for $\mathbf{L}$ must also be applied to $\mathbf{L}'$.

## 2.4 Variable localization

In addition to the spatial localization described above, we also consider the DA impact resulting from variable localization, in which potentially spurious correlations between different

model state variables are explicitly reduced. This effect can be especially important for biogeochemical models which often feature numerous state variables, most of which are typically unobserved. That is, the assimilated data contains none or very few observations to constrain most biogeochemical variables directly and variable localization is helpful to limit DA adjustments based on improper correlations in the ensemble.

We include a straightforward implementation of variable localization in our DA framework. Those correlation values associated with two different variables are multiplied by a factor of $l_{var} \in [0, 1]$, which determines the strength of variable localization. Correlation values for the same variable remain unchanged. This approach is effectively an element-wise multiplication of $\mathbf{L}$ in Eq (5) with a matrix $\mathbf{L}_{var} \in \{l_{var}, 1\}^{n_{state} \times n_{obs}}$ (and analogous treatment of $\mathbf{L}'$). To select values for $l_{var}$, we employ a similar approach as for the spatial localization and starting with a value $l_{var} = 1$, successively decreased it by half an order of magnitude, testing 0.3, 0.1, and 0.03, and selected the value resulting in the lowest prior error in 8 DA cycles. We further distinguish between physical-physical variable correlations (such as temperature-salinity correlations) and others (physical-biogeochemical and biogeochemical-biogeochemical correlations, such as temperature-$NO_3$ and $NO_3$-phytoplankton correlations, respectively). A finer-grained approach of weighting the correlations between variables based on their "dynamical distance"—for example, $NO_3$ and phytoplankton are more closely linked than $NO_3$ and zooplankton—is not tested in this study. We note right away that variable localization for correlations between the physical variables led to a degradation of results so that we set $l_{var} = 1$ for physical-physical correlations. For physical-biogeochemical and biogeochemical-biogeochemical correlations, $l_{var} = 0.1$ created the lowest prior error. We examine the effect of variable localization and different values of $l_{var}$ in more detail in Section 3.3.

## 2.5 Application to a coupled physical-biogeochemical model

In order to illustrate the capabilities of our 4dEnOI implementation, we use it to perform multiple data assimilation cycles and compare its results to that of a 4dVar benchmark implementation. As a test bed, we use a coupled physical-biogeochemical model based on the Regional Ocean Modeling System (ROMS, [34]) with a 4-variable NPZD model as the biogeochemical model component (referred to as the "NPZD iron" model in ROMS, used here with all iron-based variables deactivated). The model domain encompasses the California Current System, extending from 30˚N to 48˚N, and westward from the U.S. west coast to 134˚W (see Fig 5). This is the same model domain and physical model setup used in [5, 9, 35, 36]. The horizontal model resolution is 0.1˚ × 0.1˚; vertically, the model is split into 42 terrain-following layers. More details about the physical circulation can be found in [37].

ROMS not only contains the physical circulation model and biogeochemical model components, but also the tangent linear and adjoint code for performing 4dVar DA for the coupled model. With this, we have a test bed to assess the 4dEnOI implementation in comparison to a 4dVar benchmark implementation based on the same ROMS model.

As a benchmark, we use a strong constraint 4dVar implementation in a 2 outer loop, 10 inner loop setup which was previously presented in [5] where more details about the DA configuration can be found. In each DA cycle, it uses a static background error covariance matrix (which varies from cycle to cycle, based on the month) in which entries for unobserved variables have been lowered to prevent unrealistic DA updates. It is a log-transformed 4dVar [38], in which the chlorophyll-*a* model variable and observations are log-transformed in the computation of the 4dVar cost function and its gradients. We use this 4dVar technique as a benchmark here because we have used it in the aforementioned studies and are familiar with its configuration. Although, in the following, we assess its ability to reduce the

model-observation misfit for log-transformed chlorophyll-*a*, our reference 4dEnOI implementation does not use log-transformed biogeochemical variables internally, which is the largest difference in configuration between the two DA systems. Both 4dEnOI and 4dVar use the same cycle length of 4 days and the same prior initial conditions (in the 4dEnOI setup, this initial condition is used for the first ensemble member and to generate the initial ensemble).

## 2.6 Observations

We test both DA systems in a series of 8 cycles starting in April 2019, during which in situ and remote sensing observations of physical variables (temperature, salinity, sea level anomaly) are assimilated jointly with satellite-derived data of surface chlorophyll-*a*, all obtained from publicly available data sources (listed in Table 1). In a pre-processing step, observations were averaged, so that, at most, one observation of any observation type and data source is present in a model grid cell in each model time step (these composite observations are often referred to as super-observations, here we simply continue to use the term "observation"). In order to reduce the effect of boundary conditions on our results, observations within 10 grid cells of the model boundaries were removed from the DA. Within the 8 4-day DA cycles (using the same setup for both DA systems), a total of 232 969 observations are assimilated, just under 30 000 per cycle (Fig 3). Cloud cover frequently impedes the view of satellites, limiting the number of satellite-based observations of temperature and chlorophyll-*a*; individual grid cells are observed on average ≈4.1 times for temperature and ≈2.8 times for chlorophyll-*a* in the 32 day duration of the experiments (Fig 3a and 3b).

For observation error values, we use a 30% relative error for satellite chl *a* and 15% for in situ chlorophyll-*a*; we use reported error values (with mean value of 0.2˚C) for satellite temperature, 0.1˚C for in situ temperature; 0.05 for in situ salinity, and 2 cm for satellite SLA. These observation error values are largely based on those used in [5] for the same model, with an increase in salinity in situ error values as suggested by the results in [39] (a full calibration of observation and 4dVar background error values, as shown in [39] was not conducted).

## 3 Results

In the following, we refer to our default 4dEnOI implementation as the reference and present several modifications to the reference implementation to examine the effects of localization and other configuration choices. A full list of all 4dEnOI configurations is provided in Table 2, including short names to facilitate referencing specific experiments. We focus our description on model temperature and chlorophyll-*a*, as representative of physical and biogeochemical variables for which observations are available for assimilation.

**Table 1. The data used for assimilation.**

| observed variables | data source | URL |
| --- | --- | --- |
| SLA | Copernicus Marine Service portal | http://my.cmems-du.eu/motu-web/Motu |
| temperature (T) | NASA VIIRS | https://doi.org/10.5067/GHVRS-2PJ62 |
| chlorophyll-*a* (chl *a*) | NOAA STAR portfolio | https://www.star.nesdis.noaa.gov/portfolio |
| T + salinity (S) | ARGO floats | https://doi.org/10.17882/42182 |
| T + S + chl *a* | SPRAY Glider Network | https://spraydata.ucsd.edu |
| T + S + chl *a* | glider data from IOOS portal | https://gliders.ioos.us |

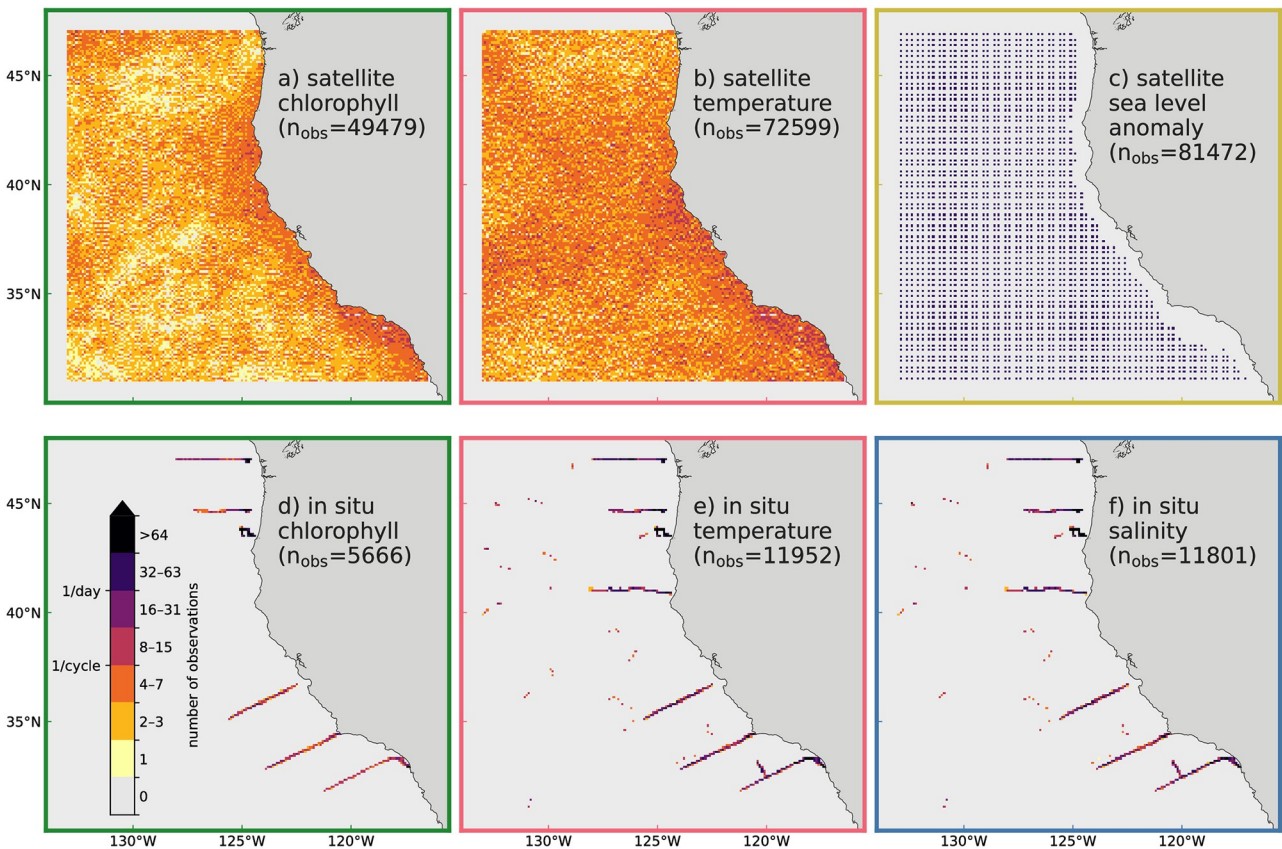

**Fig 3. Observations used for assimilation.** 2-dimensional histograms of the observations used across the 32 days (8 4-day cycles) of DA experiments. The histogram bins correspond to the horizontal model grid; satellite-based observations (top row) are only available at the surface, while the observations count for in situ observations (bottom row) includes surface and subsurface observations. Observations within 10 grid cells of the model boundaries were removed from the DA.

## 3.1 Comparison to 4dVar benchmark

In our comparison of the 4dEnOI to the 4dVar benchmark results, we first examine the reduction of the model observation misfit in both DA systems. For that purpose, we focus on the reduction in a cost function based on a weighted model-observation misfit, namely

$$J_{\text{obs}}(\mathbf{x}) = (H\mathbf{x} - \mathbf{y})^{\text{T}}\mathbf{R}^{-1}(H\mathbf{x} - \mathbf{y}) \tag{7}$$

computed for both the prior model state in each DA cycle, the prior error (also referred to as the forecast error), and the posterior model state after assimilation, the posterior error. We note that a comparison of the full cost function that is optimized in the 4dVar and 4dEnOI algorithms is not useful in this context: the full cost function contains a term which is dependent on the prior model covariance terms, which differs in both algorithms. While 4dVar and 4dEnOI thus optimize different cost functions, the main contributor to both cost functions is $J_{\text{obs}}$, which we use here as a metric of evaluation and which we hereafter refer to as cost function.

A comparison of the cost function reduction in the 8 cycles shows remarkably similar patterns for the 4dVar benchmark and our reference 4dEnOI implementation (Fig 4a). Both algorithms achieve the largest decrease of $J_{\text{obs}}$ in the first cycle, starting with initial conditions that

**Table 2. The different 4dEnOI configurations.**

| name | short description[a] | variable loc.[c] | | | | horz loc.[g] | ens. regen.[h] | | |
|---|---|---|---|---|---|---|---|---|---|
| | | sec.[b] | phy[d] | BGC[e] | vert loc.[f] | | in cycle | iterations | $\alpha_2$ [a] |
| | reference | | | ✓ | ✓ | ✓ | none | 1 | 1.0 |
| $E_1$ | not using $\mathbf{x}_1$ for statistics | 3.2 | ✓ | ✓ | ✓ | ✓ | none | 1 | 1.0 |
| $L_1$ | variable localization for all variables | 3.3 | ✓ | ✓ | ✓ | ✓ | none | 1 | 1.0 |
| $L_2$ | no variable localization | 3.3 | | | ✓ | ✓ | none | 1 | 1.0 |
| $L_3$ | no variable and no vertical localization | 3.3 | | | | ✓ | none | 1 | 1.0 |
| $L_4$ | no localization | 3.3 | | | | | none | 1 | 1.0 |
| $I_1$ | 4 iterations ($\alpha = 1$) | 3.4 | | ✓ | ✓ | ✓ | none | 4 | 1.0 |
| $I_2$ | 4 iterations ($\alpha = 0.3^{i-1}$) | 3.4 | | ✓ | ✓ | ✓ | none | 4 | $0.3^{i-1}$ |
| $I_3$ | 4 iterations ($\alpha = 0.1^{i-1}$) | 3.4 | | ✓ | ✓ | ✓ | none | 4 | $0.1^{i-1}$ |
| $G_1$ | cycle 2 regeneration | 3.5 | | ✓ | ✓ | ✓ | 2 | 1 | 1.0 |
| $G_2$ | cycle 3 regeneration | 3.5 | | ✓ | ✓ | ✓ | 3 | 1 | 1.0 |

[a] $i$ denotes the iteration index.

[b] section

[c] localization

[d] physical

[e] biogeochemical

[f] vertical localization

[g] horizontal localization

[h] ensemble regeneration

were obtained from a non-assimilative simulation. In subsequent cycles, the initial conditions are equivalent to the posterior model state at the end of the previous cycle; because these estimates have been informed by data, the prior value of the cost function is lower, and the cost function reduction is less substantial. While 4dVar and 4dEnOI achieved very similar decreases in the cost function for the first cycle, results are more variable in subsequent ones, with either of the two algorithms performing better in some cycles. Overall, the 4dVar benchmark performs slightly better in reducing the posterior error, while the 4dEnOI implementation creates marginally lower prior error values (Fig 4b).

To assess the contribution of the different observation types to the cost function, we examine the prior and posterior error for all cycles beginning with the second. By removing results from the first cycle, which is atypical because of its initial conditions, we obtain more generalizable estimates of the posterior error and a true (4-day) prior error that ignores the large prior misfits of the first cycle. The resulting cost function values, normalized by the number of observations, reveal some differences in the cost function reduction created by the 4dVar and 4dEnOI increments: while the 4dVar DA system creates a larger decrease of the model-observation misfit for SLA observations, the 4dEnOI system produces larger reductions in $J_{\text{obs}}$ for the physical and biogeochemical tracer variables, especially chlorophyll-*a* (Fig 4b).

Because both data assimilation systems start out with the same initial conditions at the start of the experiment, we can directly compare their increments in the first data assimilation cycle. For temperature and phytoplankton, which are observed variables (i.e., variables for which observations are assimilated), the two DA systems create similar surface increments, both in terms of pattern and magnitude (Fig 5). The largest differences between the increments can be found along the coast in the northern part of the domain. Furthermore, the ensemble-derived 4dEnOI increments are less smooth and more noisy than the 4dVar increments

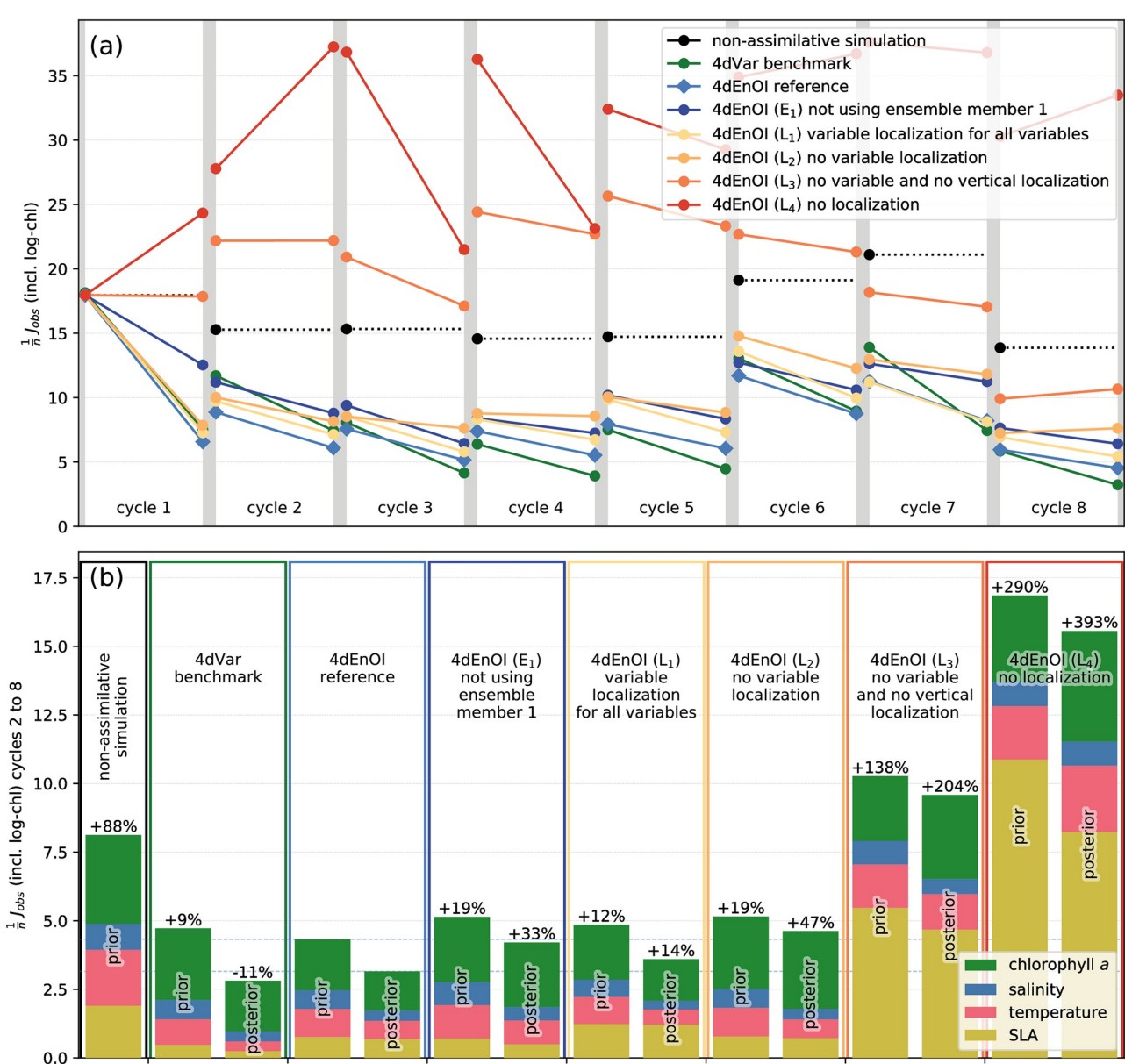

**Fig 4. 4dEnOI results for $E_1$ and different levels of localization.** (a) Prior and posterior cost function value in each of the 8 DA cycles for the 4dEnOI reference, an experiment in which the first ensemble member $\mathbf{x}_1$ is not used to generate ensemble statistics, and experiments with different levels of localization, in comparison to the 4dVar benchmark and a non-assimilative simulation using the same model. (b) Prior and posterior cost function values ($J_{obs}$ in Eq (7)) aggregated across cycles 2–8 for the same simulations. Bar segment colors indicate contribution of different observation types to the cost function values, percentage values indicate relative change of the prior or posterior cost function value to the corresponding value of the 4dEnOI reference. See Table 2 for details about each experiment.

created via tangent linear and adjoint dynamics and the choice of background error covariance values. For the observed salinity variable, increments between the two DA systems differ more, with larger magnitude and finer scale increments present in the 4dEnOI solution. The SLA increments also show notable differences. Here, it is important to note that, through thermal wind dynamics, increments in upper ocean temperature and salinity have a larger contribution towards the development of SLA in the remaining DA cycle (beyond the first model time step)

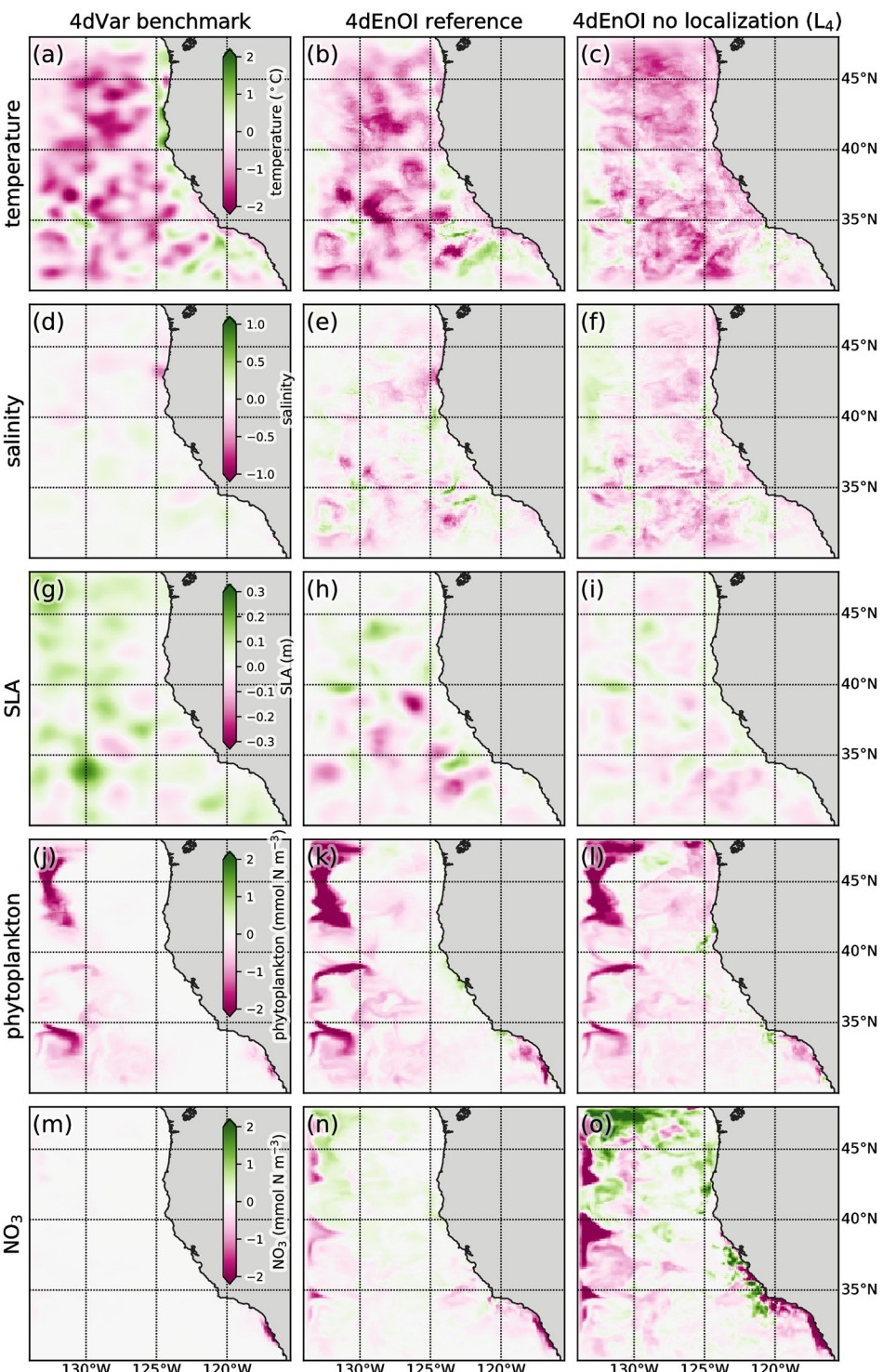

**Fig 5. Surface increments comparison.** Surface increments for the 4dVar benchmark (left column of panels), 4dEnOI reference simulation (center column), and 4dEnOI without localization (experiment $L_4$; right column) for (a-c) temperature, (d-f) salinity, (g-i) sea level anomaly, (j-l) phytoplankton, and (m-o) $NO_3$.

than the initial SLA increment itself. The lower prior and posterior error for this variable in the 4dVar system, indicates that the tangent linear and adjoint dynamical model-based SLA increment is superior to the statistical 4dEnOI increment (Fig 4b).

For the unobserved $NO_3$ variable, the 4dVar increment is considerably lower than the 4dEnOI increment. This difference is due to our configuration of both DA systems: the 4dVar implementation has lowered (static) background covariance entries for unobserved variables (see [5]), while the 4dEnOI variable localization has a large effect on the increments of unobserved variables. Without a large number of "independent" observations, it is difficult to assess which increment to the unobserved variables is more beneficial. Yet, the lower prior error values created by the 4dEnOI implementation, especially for chlorophyll-*a*, may be an indication that the increments created for the unobserved biogeochemical variables may provide beneficial dynamical feedback into the next DA cycle.

We also examined the increment of a 4dEnOI implementation identical to our reference, but without any localization (experiment $L_4$ with no spatial and no variable localization; right panels in Fig 5). Removing localization has a strong effect on the 4dEnOI increment: in many locations, the 4dEnOI-based increments do not resemble each other or show large changes in magnitude, removing localization can even flip the sign of the initial increment (for example, surface temperature increments along the coast south of 40˚N in Fig 5b and 5c). For the unobserved $NO_3$, the increment is much more pronounced without localization (compare Fig 5n and 5o). The effect of varying degrees of localization on the 4dEnOI's ability to reduce the model-observation misfit is examined in Section 3.3.

## 3.2 Using the first ensemble member for 4dEnOI statistics

In our reference implementation with a hybrid ensemble, the DA-adjusted first ensemble member, $x_1$, is used along with the remaining (static) ensemble to generate the statistics for the 4dEnOI update. Especially in comparison to using a fully static ensemble, for which ensemble statistics can be pre-computed and stored, the inclusion of the dynamically adjusted $x_1$ in the ensemble statistics adds computational cost, and prohibits their calculation before running the DA. However, initially $x_1$ is our best estimate of the model state, and after the first assimilation cycle, $x_1$ has been updated with observations. Therefore, $x_1$ could provide valuable information to the ensemble statistics. To assess the benefit of including $x_1$ in the 4dEnOI results, we perform a DA experiment, denoted $E_1$ below, in which $x_1$ is excluded from the ensemble for the purposes of computing the 4dEnOI update. This experiment is more in line with a traditional EnOI implementation with a fully static ensemble.

In our tests, the 4dEnOI reference consistently outperforms $E_1$ in all DA cycles (Fig 4a). Overall, the exclusion of $x_1$ from the ensemble statistics leads to an increase of the prior value of $J_{obs}$ by 19% and its posterior by 33% (Fig 4b). Examining the ensemble statistics more closely, the removal of the first member from the ensemble has an outsized impact compared to the removal of any other individual ensemble member in all but the first DA cycles (for an example of this general pattern, see Fig 6). The examination further reveals that the first ensemble member is less correlated to the static ensemble members than these are among themselves (not shown). Thus, including a non-static ensemble member, that has been updated with observations, appears to provide noticeably different ensemble statistics for the 4dEnOI updates, which improve the fit to the assimilated observations in our experiments.

## 3.3 Effect of localization

To assess the importance of localization for obtaining the cost function reduction presented above with an ensemble of just 25 members, we performed additional 4dEnOI experiments in

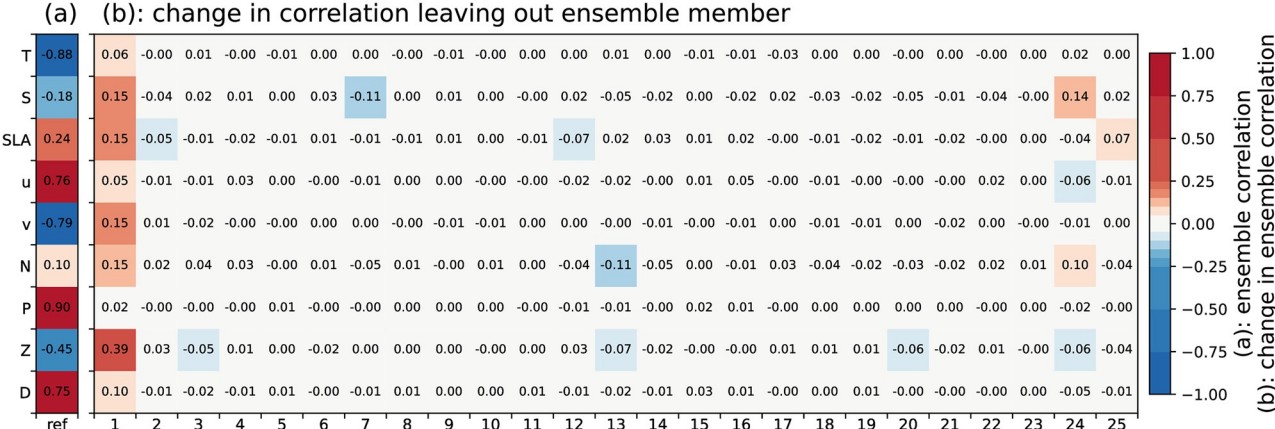

**Fig 6. Change in ensemble correlation, leaving out ensemble members.** (a) Ensemble correlation between model result at the location of a select phytoplankton (chlorophyll-*a*) observation and the prior model state at the center of the closest model grid cell. (b) Change in the ensemble correlation when a single ensemble member (1 to 25) are left out of the ensemble. The observation is a surface observation at 41.62˚N, 128.98˚W, 3 days into the second data assimilation cycle (2019–04-08 12:30 UTC); no localization is applied.

which localization is progressively modified, but which are otherwise identical to the reference 4dEnOI experiment. Experiments associated with different localization configurations are denoted $L_1$ to $L_4$ in Table 2. In the first two experiments, we consider variable localization, applying variable localization to all variables, including the physical-physical covariance values in experiment $L_1$, and in experiment $L_2$ removing variable localization from the DA system altogether. In terms of spatial localization, experiments $L_1$ and $L_2$ are identical to the reference experiment. In experiment $L_3$, we remove all variable localization as in $L_2$ and also exclude vertical localization. The final experiment $L_4$, removes all variable and spatial localization completely (i.e., it uses no localization). In our experiments, we use relatively strong variable localization with $l_{var} = 0.1$ when variable localization is applied (after testing values of $l_{var} = 0.03, 0.1, 0.3$); whereas no variable localization is equivalent to $l_{var} = 1$ (Section 2.4).

With variable localization applied to all variables in $L_1$, the prior and posterior errors increase by a few percent from the 4dEnOI reference (Fig 4b). Proportionally, the largest increase in the cost function occurs for SLA observations, highlighting the importance of adjustments of temperature and salinity (and hence interior density) to correct sea surface height. Thus, adding the same level of variable localization that is found to be beneficial for physical-biogeochemical covariances to the SLA-temperature and SLA-salinity covariances has a (modestly) detrimental impact. Overall, however, variable localization has a positive effect on the cost function reduction: in $L_2$, without any variable localization, prior and posterior cost function values increase further, with an almost 50% higher posterior $J_{obs}$ compared to the reference simulation.

While the 4dEnOI algorithm can still create significant cost function reductions without variable localization, if both variable and vertical localization are removed ($L_3$), most 4dEnOI increments become detrimental and $J_{obs}$ increases to values higher than in the non-assimilative simulation. The model-observation misfit increases even more in $L_4$, which uses no localization whatsoever, emphasizing the importance of localization for obtaining improved state estimates from small ensembles.

To assess the importance of spurious correlations in this application, we examine the correlations between the model ensemble at the location of a representative observation and the ensemble of initial conditions **X** (that is, the entries of one row of **L** ∘ cov (**X**, *H* **X**) scaled to

correlation values). Specifically, we compare the correlations in the 4dEnOI reference experiment which uses spatial and variable localization to experiment $L_4$ without any localization (Table 2). Without localization, correlations of large magnitude extend across the entire model domain, even spatially far from the observation location and for variables different from the one observed (left column in Fig 7); they also extend below the surface (not shown). Localization effectively reduces the correlation values that are spatially distant from the observation and—through variable localization—for variables other than the one observed (right column in Fig 7).

To briefly examine the influence of an increase in ensemble rank on the 4dEnOI results, we compute the condition number for the matrix that is being inverted in Eq (1). In the first DA cycle of the reference simulation, the $29090 \times 29090$ matrix has a condition number of $\kappa_{\|\cdot\|_2}(\mathbf{L}' \circ \mathrm{cov}(H\mathbf{X}, H\mathbf{X}) + \mathbf{R}) = 122\,872$. When successively removing localization, the condition number changes to $124\,335$ (for no variable localization $L_2$), $127\,605$ (no variable or vertical localization $L_3$), and $1\,461\,520$ (no localization $L_4$), increasing by more than an order of magnitude. We conclude that localization has a positive impact on the ensemble rank in this application.

## 3.4 Multiple EnOI iterations

The 4dVar benchmark implementation performs DA in two so-called outer loops. Each outer loop begins a new (nonlinear) model simulation, the first of which is unperturbed using the prior initial conditions and each subsequent one adding the increment obtained from the preceding outer loop. These nonlinear model simulations provide the model state about which the tangent linear calculation is linearized, and each is used to compute a new increment (see, e.g., [40] for more information).

In the 4dEnOI implementation, we can include a procedure analogous to 4dVar outer loops: we update the first ensemble member repeatedly with a 4dEnOI increments in a series of iterations. The first iteration is performed like the regular 4dEnOI DA procedure described above, producing an increment to update the initial state estimate. Each following iteration updates the previously updated state estimate. This step requires re-running the model with the previously updated state $\mathbf{x}_1^*$, in order to obtain $H\mathbf{x}_1^*$, which is then used to update the state again using Eq (1). Each iteration thus uses a different initial state estimate and slightly modified ensemble statistics, because the first ensemble member has changed. The computational cost for each new iteration (consisting of a model simulation and an 4dEnOI update), is the same as that of the first iteration when a static or hybrid ensemble is used.

We test this procedure in experiment $I_1$, which uses the 4dEnOI reference configuration without any modifications, but performs four iterations in each cycle. The increments in the first iterations create a substantial cost function reduction and are identical to those obtained for the 4dEnOI reference (dark purple lines in Fig 8a). In most cycles, the second increment further decreases the cost function, but in some cycles the cost function increases. This effect intensifies in subsequent iterations, with more and larger cost value increases in iterations three and four, leading to an overall cost function increase (+4% in the prior and +17% in the posterior cost function value). We presume that these increases are caused by increased sensitivities to errors in the ensemble-based covariance estimates due to smaller values in the model-observation difference $\mathbf{y} - H\mathbf{x}$. In order to reduce this effect, we decrease the value of the scaling factor $\alpha$ used to reduce the 4dEnOI increment (see Eq (1)) in each iteration. More precisely, we set $\alpha = \alpha_2^{i-1}$ (i.e., $\alpha_2$ raised to the $i-1$ power), where $i = 1, 2, 3, 4$ is the iteration index, so that $\alpha$ remains 1 in the first iteration, and $\alpha_2$ is the value of $\alpha$ in the second iteration. We perform two experiments: experiment $I_2$ with $\alpha_2 = 0.3$ and experiment $I_3$ with $\alpha_2 = 0.1$. In

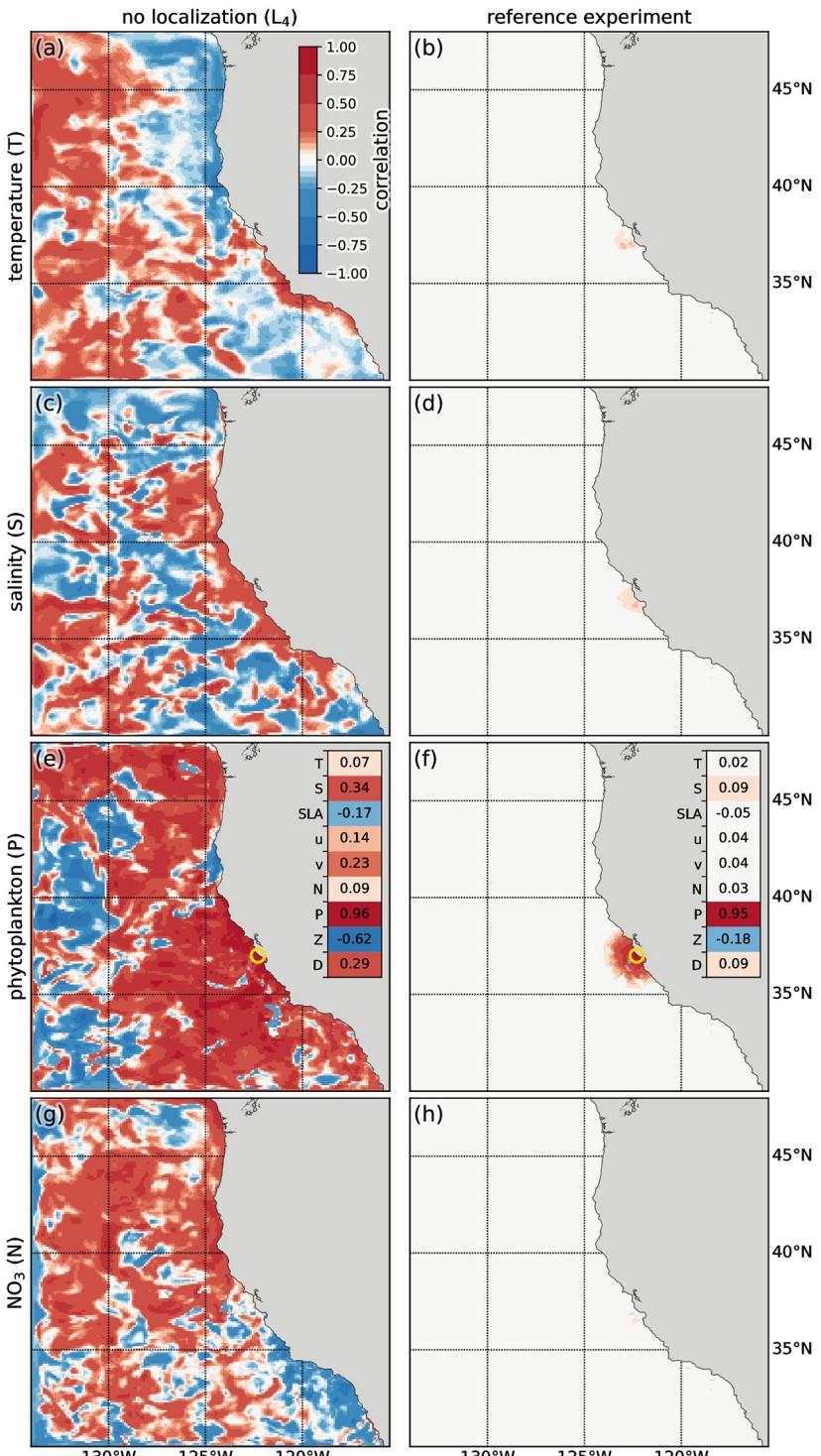

**Fig 7. Correlation structure.** Correlation between the ensemble of initial conditions and a representative chlorophyll-*a* (phytoplankton) observation in the 4dEnOI without any localization (left panels) and the 4dEnOI reference with localization (right panels). Correlation values show surface (a, b) temperature, (c, d) salinity, (e, f) phytoplankton, and (g, h) NO₃ in the first DA cycle, the observation location is marked by the green ring in (e and f). Insets (e and f) show the correlation of the same observation to the initial conditions for all 9 state variables, at the spatial location of the grid cell closest to the observation location.

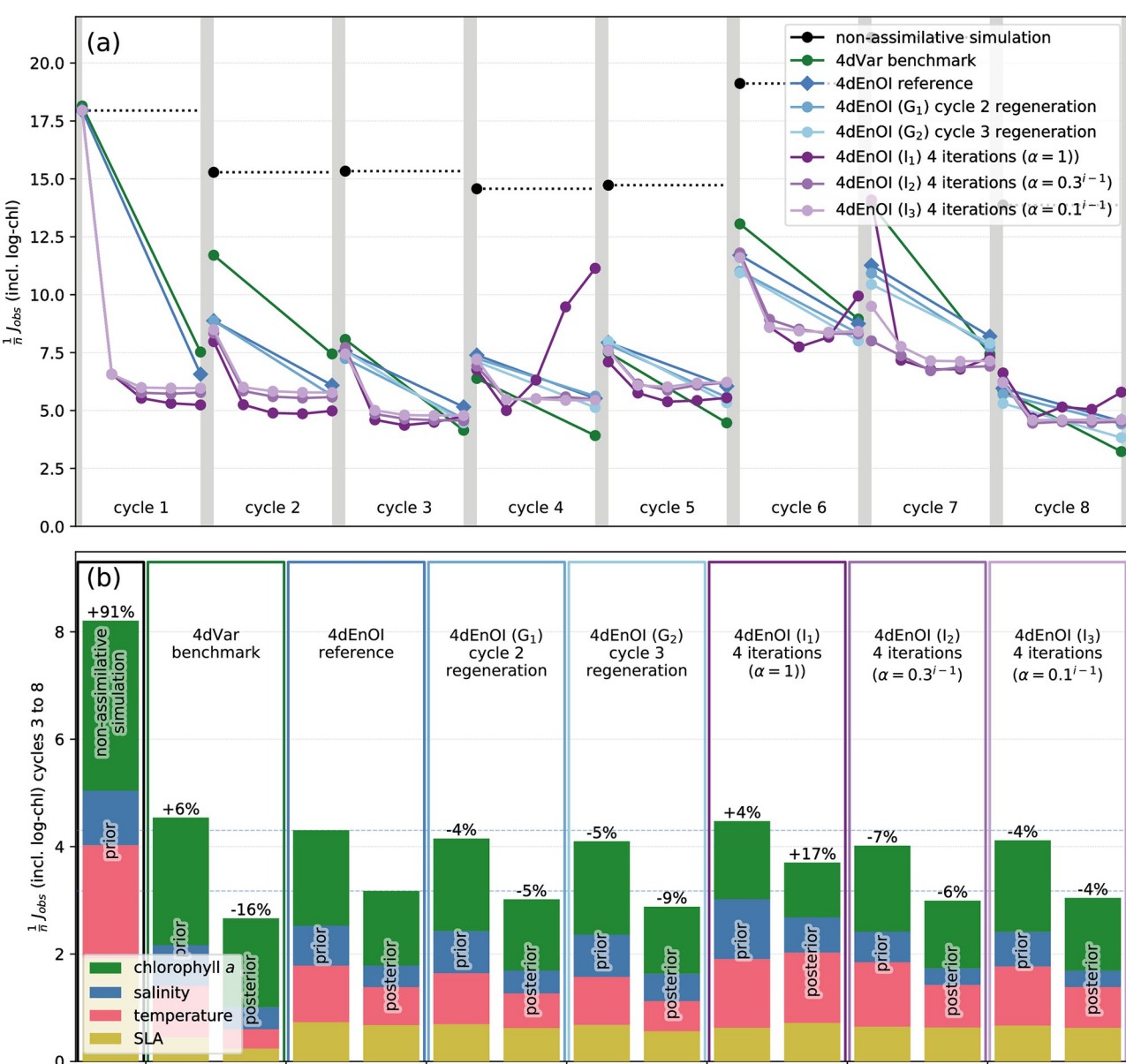

**Fig 8. 4dEnOI results for multiple iterations and ensemble regeneration.** (a) Prior and posterior cost function value in each of the 8 DA cycles for 4dEnOI experiments with multiple iterations (purple; see Section 3.4) and ensemble regeneration (blue; Section 3.5) in comparison to the 4dEnOI reference, 4dVar benchmark, and non-assimilative simulation. (b) Prior and posterior cost function values aggregated across cycles 3–8 for the same simulations. Bar segment colors indicate contribution of different observation types to the cost function values, percentage values indicate relative change of the prior or posterior cost function value to the corresponding value of the 4dEnOI reference. Cost function values for 4dEnOI reference, 4dVar benchmark, and non-assimilative simulation are identical to those in Fig 4, but do not include cycle 2 in the aggregate because the "cycle 3 regeneration" experiment was started in cycle 3.

experiment $I_2$, the 4dEnOI algorithm reduces its increments and creates only a few, less substantial cost function increases in later iterations of a few cycles. It creates an aggregated cost function decrease −7% for the prior and −6% for the posterior cost function value with respect to the reference experiment with just one iteration. For $\alpha_2 = 0.1$ in experiment $I_3$, the 4dEnOI increments are reduced to such an extent that they have little impact beyond the first iteration

of each cycle, and the aggregated cost function reduction is less substantial (prior: −4%, posterior: −4%).

Especially in the face of the relatively modest improvement gained from multiple 4dEnOI iterations, it is important to note that every additional iteration can be associated with substantial computational cost. In our current, naive implementation, each new iteration has the same computational cost as the first one, so that a 4dEnOI assimilation step with two iterations takes roughly twice as long as a single iteration setup.

### 3.5 Ensemble regeneration

While the first ensemble member is updated with observations, the remaining, static, 4dEnOI ensemble was generated from the first ensemble member before any data is assimilated (see Section 2.2) and remains static, without any further modifications from data. While a static ensemble has computational benefits (the variable localization experiments or the multiple iteration experiments above can be performed without the need to rerun the full ensemble), the results from Section 3.2 indicate that an ensemble with ensemble members closer to the observations can improve the ensemble-based statistics and yield better state estimates. An obvious approach to keep the remaining ensemble distributed around the first ensemble member which has been updated with previous observations, is to regenerate the ensemble from the first ensemble member after a number of DA cycles have been performed. Such an approach eliminates one of the main computational benefits of the 4dEnOI procedure, but we use it here to examine the effects of using a naive ensemble, uninformed by observations, to one that has been passively updated with past observations using ensemble regeneration after the first or second DA cycle.

We examine the effect of ensemble regeneration by reconstructing the ensemble of our 4dEnOI reference simulation in DA cycles 2 and 3, after the largest DA update has occurred. That is, in the first experiment $G_1$, we use the prior initial condition from cycle 2 of the reference experiment (updated by observations assimilated in cycle 1) to create a new ensemble that is then used to perform 4dEnOI DA for cycles 2 to 8 using the reference configuration (the same localization configuration, and a single 4dEnOI iteration). In the second experiment, $G_2$, a new ensemble is generated from the prior initial condition in cycle 3 of the reference simulation, which is then run forward through cycles 3 to 8. Both experiments improve the 4dEnOI results and reduce the prior and posterior values of the cost function by ≈ 5% compared to the reference simulation (Fig 8). Thus, it appears useful, but not necessary, for good DA results to use a static ensemble closer to the observations—or to obtain such an ensemble by regenerating the ensemble from a data-assimilative state. In this application, the first cycle is associated with the largest increment and the largest move of the first ensemble member away from the center of the ensemble. Yet, the ensemble is still suitable to perform DA and the benefits of ensemble regeneration remain modest. There is little change between the results of $G_1$ and $G_2$, indicating that the timing of ensemble regeneration is not critical. Yet, we presume that ensemble regeneration would provide more benefits in other applications when the ensemble has drifted away from the first member, in cases of ensemble collapse, or after more DA cycles.

The process of regenerating the ensemble is associated with additional computational cost. By far, the largest expense is associated with running the full ensemble forward instead of relying on a static ensemble. Additional resources are required for generating the principal components (PCs) in the ensemble regeneration process (Fig 2). Fortunately, the PCs can be precomputed and stored in the initial ensemble generation step, and the remaining computation of weights and multiplication of perturbed weights with the PCs (Fig 2) is significantly

less costly than a 4dEnOI iteration ($<$ 20% in terms of computer runtime in our naive implementation with $n_{pc}$ = 25 PCs and $n_{ens}$ = 25 ensemble members).

## 4 Discussion & conclusions

The three main goals of this study were to develop a DA technique that is computationally practical, easy to implement, and provides good DA performance for complex biogeochemical models with $>$ 30 state variables and consequently a large model state size. The 4dEnOI technique presented here fulfills these goals. (1) The computational cost of the technique increases linearly with the size of the model state, while the memory requirement is adjustable and does not grow with the number of state variables. It permits the use of a static ensemble, reducing the cost of repeat DA experiments, for example, for finding suitable localization length scales or weights ($l_{var}$). (2) It is easy to implement and, relying only on ensemble-based statistics, requires no tangent linear or adjoint code. (3) For the relatively simple biogeochemical NPZD model with 4 state variables, it provides similar performance to our reference 4dVar benchmark implementation, both in terms of the analysis and forecast model error reduction.

The technique that we label "4dEnOI" in this study lives somewhere on the spectrum between the original EnOI and variations of the EnKF technique. When EnOI was first introduced in [19], its main distinguishing features—compared to the original EnKF—were the use of a static ensemble and a single state vector that is updated by DA (in contrast to an ensemble in the EnKF). Various studies have used EnOI implementations in the context of static ensembles [41, 42], and a few have added more flow-dependent statistics by using time-varying static ensembles (i.e., using a different static ensemble for each DA cycle [20, 31]). Thus, [43] point out that "using a seasonal or even dynamic ensemble still qualifies the method as the EnOI, as long as there is no feedback from DA to the background covariance". Our implementation stretches this definition a bit further by including the assimilative first ensemble member in an otherwise static ensemble that is not updated by the DA. The resulting technique thus creates a hybrid background error covariance matrix generated from a hybrid ensemble—a mostly static ensemble augmented by a (single-member) dynamical one—analogous to EnKF implementations that use hybrid covariances (see, e.g., [29]). The use of a (mostly) static ensemble in this application requires storing the initial conditions at the start of each assimilation cycle (model state snapshots every 4 days), along with model results at the observation locations, for $n_{ens} - 1$ = 24 simulations. Furthermore, indirect updates occur in the optional ensemble regeneration process (see Section 3.5), when the first ensemble member is used to regenerate the ensemble around an updated model state. But because the remaining ensemble remains static after a regeneration, and because there is no attempt to track the second moment of the model state distribution (the covariance matrix) from cycle to cycle, we decided to use the EnOI name for this technique. We prepended a "4d" to the name, to signify the use of asynchronous DA, here implemented using a hybrid ensemble.

The use of dynamical, flow dependent statistics without linearized observation operators is one of the distinguishing features of our 4dEnOI implementation. Other studies applying DA to ocean models have compared EnOI implementations with different levels of flow-dependence to the EnKF: in a study using a static ensemble for the EnOI, the EnKF performed better [43]; when more flow-dependence is added by generating static ensembles from a floating temporal window, the EnOI compares well to the EnKF, both producing similar results [31]. In this study, we use fully flow-dependent statistics for our EnOI implementation, and compare it to a 4dVar implementation. Due to the small ensemble, noise is notable in the increment that is created by the 4dEnOI update, and the technique cannot reduce the model error for SLA observations as much as the 4dVar benchmark. However, the 4dEnOI implementation

performs better at reducing the error for some observation types, especially chlorophyll-*a*, and, overall, the 4dEnOI and 4dVar implementations create very similar reductions in the cost function. In our experiments, the choice of 25 ensemble members was mainly based on the computing resources available to us. We presume that a larger ensemble would improve the 4dEnOI results, while also reducing the relevance of including the first, and only non-static, ensemble member in the computation of the ensemble statistics (see Section 3.2).

Like in other ensemble-based DA studies, localization plays an important role in creating beneficial state increments. Our results indicate that spatial localization and, to a lesser extent, biogeochemical variable localization are essential for obtaining the cost function reduction with an ensemble of just 25 members. Without any localization, the 4dEnOI update led to an increase in the cost function in our results, worsening the model-observation misfit compared to a non-assimilative simulation. In our experiments, horizontal localization proved to be the most important, and was required to improve the model fit to observations—additional vertical localization further improved the results. We also tested a simple variable localization scheme which, when combined with the spatial localization, provides the best results in this study. That is, retaining the strong coupling between the physical model variables, while reducing physical-biogeochemical and biogeochemical-biogeochemical covariance terms yielded the lowest model-observation misfit in our experiments. This result suggests that the correlations of less related—dynamically distant—variables are overestimated by the ensemble, analogous to correlations between spatially distant grid points. Beyond decreasing the model-observation misfit, variable localization likely also reduces unrealistic increments to unobserved biogeochemical variables, a positive effect that is difficult to measure due to the lack of observations for these variables.

While we tested different horizontal and vertical length scales for our 4dEnOI implementation in preparation for this study, we note here that we did not examine a variety of potentially useful localization configurations. For example, the 4dVar benchmark uses a different vertical length scale for physical and biogeochemical variables [38], a feature we did not test in the 4dEnOI implementation. Similarly, we tested a few variable localization strengths before our experiments, but did not examine the effect of a more fine-grained approach, applying different variable localization strengths to different variable combinations, beyond distinguishing between physical and biogeochemical variables. For example, a typical NPZD biogeochemical model simulates a nutrient → phytoplankton → zooplankton → detritus → nutrient cycle, and the results in Section 3.3 indicate that there may be benefit in applying weaker variable localization to more closely related, dynamically less distant, variables, such as nutrient-phytoplankton covariance terms, compared to nutrient-zooplankton covariances.

The main reason for using a NPZD model in our study, was the availability of a 4dVar implementation for this model. The 4-variable model further offers a simple framework to examine the coupling between the variables (including physical ones) and the effect of variable localization. In a more complex biogeochemical model, interactions between the variables tend to become more complex, for example, the generic phytoplankton variable in an NPZD model is typically represented by multiple variables representing different phytoplankton functional types. Besides challenges of how to assimilate satellite-derived chlorophyll-*a* with multiple phytoplankton variables (see, e.g., [6] for a solution), it becomes more difficult to judge the dynamical distance or linkage strength between the different variables and assess whether ensemble-based estimates of correlations between variables are spurious. Additionally, in more complex biogeochemical models, fewer variables are observed, i.e. directly constrained by the observations that are assimilated. This circumstance makes it more difficult for DA techniques to perform beneficial state increments, and for modelers to evaluate the quality of

increments and thus the DA performance. The large number of unobserved variables may be the largest problem facing DA for complex biogeochemical models. Here, the 4dEnOI technique provides no simple solution to this problem, but the means to quickly examine variable correlations and perform variable localization experiments to limit the effect of spurious correlations between biogeochemical variables.

The aforementioned problem of unobserved biogeochemical variables also affects variational DA techniques: in previous experiments, the user-prescribed 4dVar background error covariance terms had an outsized influence on the biogeochemical increment of unobserved variables [5]. For example, an increase in unobserved zooplankton or a decrease in unobserved $NO_3$ can lead to a similar decrease in the observed phytoplankton, and the 4dVar background error covariance terms largely determine which variable is incremented. The lack of covariance terms between different state variables in the background error covariance matrix in the ROMS 4dVar implementation can be a further cause of unrealistic increments to unobserved variables. Thus, increments to unobserved variables in our 4dVar implementation remain difficult to judge objectively without suitable data. Fortunately, going forward, numerous emerging ocean observation platforms [44] promise novels datasets for informing increasingly complex biogeochemical models.

Many studies mentioning the computational requirements of the EnOI technique, emphasize the need for only a single model simulation as its main computational benefit [17, 42, 45]. In practice, we find that it is very easy to parallelize model simulations across different cluster computer nodes, so that most, if not all, model simulations can easily be performed simultaneously, and an ensemble regeneration is not out of the question. In contrast, it is more difficult to parallelize the 4dEnOI update across different nodes (as opposed to the CPU cores of a single node). The main reasons for the comparatively slow 4dEnOI update in our implementation, is that we have so far spent little effort in optimizing its code. Thanks to the splitting of the model state (see S1 File), it is not difficult to speed the update, for example, by computing the updates for different parts of the model domain in parallel.

With the advantages mentioned above, the 4dEnOI implementation presented here can be used operationally. We see two main areas of applications: Firstly, DA for models for which no tangent-linear and adjoint models exist and are difficult to create, such as—the main motivation for this study—complex biogeochemical models. Secondly, given the importance of spatial and variable localization, and the need to parameterize the localization techniques with length scales and similar parameters, the 4dEnOI lends itself to finding suitable values for these parameters at a modest computational cost. Our results suggest that, with proper calibration, 4dEnOI can be competitive with 4dVar implementations. However, if tangent linear and adjoint models already exist, it may be more fruitful to calibrate the 4dVar implementation than implement a simple ensemble-based DA technique. If the ultimate goal is to employ an EnKF or similar more complex ensemble-based DA technique, the 4dEnOI may be used to tune localization parameters first, which can then be transferred to the more complex technique. In such a scenario, the 4dEnOI implementation could further act as a benchmark for DA performance.

As a next step, we are planning to apply the 4dEnOI technique to a biogeochemical model with more than 30 variables and a small ensemble size. The 4dEnOI technique permits us to implement DA quickly and calibrate the implementation efficiently. For small ensembles, spurious correlations can affect dynamically distant variables in the same way as spatially distant grid points. This circumstance emphasizes the need for a carefully calibrated variable localization, especially for models in which many variables are unobserved and dynamically distant from observed variables.

## Supporting information

**S1 File.**
(PDF)

## Acknowledgments

We thank four anonymous reviewers for their constructive comments and helpful suggestions.

## Author Contributions

**Conceptualization:** Jann Paul Mattern, Christopher A. Edwards.

**Formal analysis:** Jann Paul Mattern.

**Funding acquisition:** Christopher A. Edwards.

**Investigation:** Jann Paul Mattern.

**Methodology:** Jann Paul Mattern, Christopher A. Edwards.

**Supervision:** Christopher A. Edwards.

**Validation:** Jann Paul Mattern.

**Writing – original draft:** Jann Paul Mattern.

**Writing – review & editing:** Christopher A. Edwards.

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
