## [Decision Letter · Decision Letter 0]

30 May 2022

PONE-D-22-03604

Ensemble optimal interpolation for adjoint-free biogeochemical data assimilation

PLOS ONE

Dear Dr. Mattern,

Thank you for submitting your manuscript to PLOS ONE. After careful consideration, we have decided that your manuscript does not meet our criteria for publication and must therefore be rejected.

Specifically:

I am sorry that we cannot be more positive on this occasion, but hope that you appreciate the reasons for this decision.

Kind regards,

Ibrahim Hoteit

Academic Editor

PLOS ONE

Additional Editor Comments:

I am now in receipt of comments from the three reviewers. I overall agree with the reviewers comments that the proposed methods not only lack mathematical rigorous but their rationales are not explained nor well justified. As also suggested by the third reviewers, the analysis of the results is also not complete. I am sorry I cannot be more positive at this level. I hope that the reviewers comments will still be helpful for the authors.

Reviewers' comments:

Reviewer's Responses to Questions

**Comments to the Author**

1. Is the manuscript technically sound, and do the data support the conclusions?

Reviewer #1: Yes

Reviewer #2: No

Reviewer #3: Partly

2. Has the statistical analysis been performed appropriately and rigorously? 

Reviewer #1: Yes

Reviewer #2: N/A

Reviewer #3: N/A

3. Have the authors made all data underlying the findings in their manuscript fully available?

Reviewer #1: No

Reviewer #2: Yes

Reviewer #3: Yes

4. Is the manuscript presented in an intelligible fashion and written in standard English?

Reviewer #1: Yes

Reviewer #2: No

Reviewer #3: Yes

5. Review Comments to the Author

Reviewer #1: Title: Ensemble optimal interpolation for adjoint-free biogeochemical data assimilation

Authors: Mattern and Edwards

Recommendation:

Summary

This manuscript proposed a four-dimensional ensemble optimal interpolation (4dEnOI) for a simple marine ecosystem model. The 4dEnOI uses regenerated ensemble ICs to launch ensemble forecasts and incorporate flow-dependent background error covariances. Various localizations, including variable localization, horizontal and vertical localization, are examined. Results show that 4dEnOI performs similarly to 4DVAR. The 4dEnOI highly depends on covariance localization, and it can be improved with multiple iterations. The manuscript is well written, but there are places where clarifications and explanations are needed. Please see my comments below.

1. It is unclear how the algorithm proposed here is a four-dimensional one. It mentioned that “observations that vary both in space and time (and are thus 4-dimensional)”. But if the observations within an assimilation window is assumed to happen at the assimilation time, or a FGAT is used, it is not really a four-dimensional method. Are multiple time slices of state variables used in an assimilation window?

2. The authors state that “What sets our implementation further apart from other 4dEnOI approaches is that it does not use a static ensemble of model simulations. Instead, it is based on flow-dependent statistics which are obtained from a dynamical ensemble using spatial and variable localization to obtain more reliable statistics.” As described in section 2.2, for each assimilation cycle, the initial ensemble ICs are generated by projecting the updated x1 on climatological PCAs with perturbations. Then the ensemble ICs are integrated forward. The climatological PCAs approximates the leading components of a static background error covariance, thus the forecasted ensembles approximate the static background error covariance, rather than the common EnKF’s flow-dependent background error covariance. Moreover, ensemble model integration requires much more computational cost than the assimilation. Since an ensemble model integration is used, why not just use the common EnKF? One more comment is whether ensemble mean equals x1? If not, the ensemble perturbations are biased.

3. Are the horizontal localization of 1 degree and vertical localization of 300 m optimal? Since an exponential localization function (Eq. 8) is used, the observation impact does not go to the exact 0 at a certain distance?

4. Section 2.5, is the parallelization described here similar to Anderson and Collins 2007 (Scalable implementations of ensemble filter algorithms for data assimilation)?

5. Figure 3b, it is interesting that 4dEnOI has smaller errors than 4DVAR. 4DVAR has temporal propagation of background error covariance matrix and localization matrix during an assimilation window, which should be beneficial. Please explain why 4dEnOI has advantages over 4DVAR here?

6. l365-374, with or without variable localization, there are no significantly different performances for 4dEnOI. Intuitively, variable localization could exacerbate the imbalance of analysis increment and subsequent model forecast. To quantify the imbalance, can the surface pressure be verified?

7. Section 3.4, with multiple iterations, how’s the computational cost increase for 4dEnOI?

Minor comments

1. l9, please add some references for ‘variational DA techniques’.

2. l13, please add some references for ‘ensemble-based DA’.

3. l95, X should have dimension of ‘n_state * n_ens’.

4. l98, HX should have dimension of ‘n_obs * n_ens’.

Reviewer #2: Ensemble Optimal Interpolation for Adjoint-Free Biogeochemical Data Assimilation

Jann Paul Mattern and Christopher A. Edwards

The authors present a biogeochemical ocean data assimilation (DA) study in the California Current System. The DA framework uses a variant of the ensemble optimal interpolation (EnOI) scheme with several modifications to the original EnOI algorithm. In particular, the static background covariance is not static anymore, a variable localization is utilized and furthermore a scaling factor is used to limit the impact during the update. The authors also propose a way to regenerate the ensemble in case the first member starts drifting away from the rest of the ensemble. I find the new method quite ad-hoc with very little mathematical rigor and limited scientific impact. The authors however do compare its performance to an already existing 4DVAR solution. The results show that the proposed 4dEnOI system can produce comparable performance to 4DVAR. I recommend rejection of the article.

1- I am struggling to find a motivation for the new EnOI method. We know for a fact that an EnKF with a full dynamic covariance outperforms the EnOI with a static (and sometimes seasonal) covariance. The whole motivation for using EnOI is to cut down on the massive computational burden that one pays by running a full ensemble. The authors are in fact integrating the entire ensemble using the model so why not simply use the EnKF? We know that it gives better estimates than a single-member EnOI. I find this quite troubling. I understand that the simple NPZD-type system is quite cheap to integrate forward in time, but this is rarely the case in large and more realistic models.

2- My second major take on the method is the issue of only updating the first member. I find it difficult to justify this choice. If the entire ensemble is dynamic, why not update all the members. I can see scenarios in large models where only updating the first member can cause inconsistencies leading to divergence and model instabilities. The authors further suggest an ensemble regeneration technique where the new members are sampled around the posterior (i.e., the first member). This feels counter-productive adding an extra layer of complexity to an already complex framework. On the other hand, the EnKF gives you an ensemble after each assimilation cycle that is centered around the posterior estimate which happens to be the mean.

3- It seems to me that a lot of tuning is required for the method to become competitive. Tuning includes optimizing the scaling factor alpha (also known as damping factor in the literature e.g., Hendricks Franssen and Kinzelbach 2008), finding the right variable localization and deciding when to regenerate the ensemble. This makes the technique less desirable in other applications.

4- As far as variable localization goes, what is a good strategy to select lvar? Ideally, using a statistically sound ensemble lvar should be set to 1 and it is left to the model correlations to determine the impact of the update on the variables in the state. I’m just thinking in a large state where you have tens of variables tuning this variable localization would be a hassle.

5- I suggest rewriting section 2. Many points need to be summarized and perhaps removed and citations to relevant literature could be added instead. For instance, ensemble generation using PCA by projecting onto the leading modes is well-known and has been introduced and discussed extensively in the literature. The first paragraph in Section 2.3 about localization can also be omitted. After more than 20 years, DA readers should be quite familiar with localization and the need for it. Section 2.5 describes a splitting strategy to make the Kalman update more efficient. Operational centers and many DA software have been using different ways to make the update more efficient. Splitting is something that has been used in many other systems. For instance, the TOPAZ system (Sakov et al., 2012) splits/slices the update by vertical layers.

Other comments:

1- Abstract: Computationally suitable instead of suitable computationally.

2- Abstract: its results instead it results?

3- Line 55: and *is able to* accommodate changes.

4- Line 68: variable localization is vague here. It could be described briefly here.

5- Line 98: Shouldn’t HX be Nobs x Nens?

6- Line 105: Add a reference to motivate the use of alpha.

7- Equations 3 and 4: I don’t see the point of this approximation. Should these statistically be equivalent even though B is time-invariant? Adding time subscripts in your equations may help.

8- Line 308: reveal instead of reveals.

9- Section 3.4: I don’t quite understand the iterative procedure. Again, adding equations here can be very helpful. I though you’re iterating over the update equation at a single time. If you have to rerun the model starting from the posterior, doesn’t that take you to the next cycle?

10- Line 493: 4dEnOI or 4DEnOI. It’s good to stick to one notation.

Reviewer #3: I apologize for the delay of my report. Please find my comments in attachments.

6. PLOS authors have the option to publish the peer review history of their article (what does this mean?). If published, this will include your full peer review and any attached files.

Reviewer #1: No

Reviewer #2: No

Reviewer #3: No

- - - - -

---

## [Author Response · Author response to Decision Letter 0]

15 Dec 2022

We have included all comments and responses in the uploaded document.

---

## [Decision Letter · Decision Letter 1]

28 Mar 2023

PONE-D-22-03604R1

Ensemble optimal interpolation for adjoint-free biogeochemical data assimilation

PLOS ONE

Dear Dr. Mattern,

Thank you for submitting your manuscript to PLOS ONE. After careful consideration, we feel that it has merit but does not fully meet PLOS ONE’s publication criteria as it currently stands. Therefore, we invite you to submit a revised version of the manuscript that addresses the points raised during the review process.

In particular, as you will see from the Reviewers' reports, both the reviewers and myself believe that some specific choices of the data assimilation systems and experimental results can be better emphasized and discussed in the manuscript.

We look forward to receiving your revised manuscript.

Kind regards,

Andrea Storto

Academic Editor

PLOS ONE

“This research was supported by the Simons Foundation (CBIOMES award ID: 549949)”

“JPM and CAE were both funded by the Simons Foundation (CBIOMES award ID: 549949).

https://cbiomes.org/

3. We note that Figure 6 in your submission contain [map/satellite] images which may be copyrighted. All PLOS content is published under the Creative Commons Attribution License (CC BY 4.0), which means that the manuscript, images, and Supporting Information files will be freely available online, and any third party is permitted to access, download, copy, distribute, and use these materials in any way, even commercially, with proper attribution. For these reasons, we cannot publish previously copyrighted maps or satellite images created using proprietary data, such as Google software (Google Maps, Street View, and Earth). For more information, see our copyright guidelines: http://journals.plos.org/plosone/s/licenses-and-copyright.

1. You may seek permission from the original copyright holder of Figure(s) [#] to publish the content specifically under the CC BY 4.0 license.

3. We note that your manuscript is not formatted using one of PLOS ONE’s accepted file types. Please reattach your manuscript as one of the following file types: .doc, .docx, .rtf, or .tex (accompanied by a .pdf).

If your submission was prepared in LaTex, please submit your manuscript file in PDF format and attach your .tex file as “other.”

4. Please upload a copy of Figures 2 and 4 which you refer to in your text on pages 5 and 9. Or if the figure is no longer to be included as part of the submission please remove all reference to it within the text.

Additional Editor Comments (if provided):

Reviewers' comments:

Reviewer's Responses to Questions

**Comments to the Author**

1. If the authors have adequately addressed your comments raised in a previous round of review and you feel that this manuscript is now acceptable for publication, you may indicate that here to bypass the “Comments to the Author” section, enter your conflict of interest statement in the “Confidential to Editor” section, and submit your "Accept" recommendation.

Reviewer #1: All comments have been addressed

Reviewer #4: (No Response)

2. Is the manuscript technically sound, and do the data support the conclusions?

Reviewer #1: Yes

Reviewer #4: Yes

3. Has the statistical analysis been performed appropriately and rigorously? 

Reviewer #1: Yes

Reviewer #4: Yes

4. Have the authors made all data underlying the findings in their manuscript fully available?

Reviewer #1: Yes

Reviewer #4: Yes

5. Is the manuscript presented in an intelligible fashion and written in standard English?

Reviewer #1: Yes

Reviewer #4: Yes

6. Review Comments to the Author

Reviewer #1: Thank the authors for their efforts to address previous comments. The manuscript has been significantly improved. I have a few comments as below.

1. Since the spatial localization function (6) does not go to zero exactly, any threshold is applied to force it being zero at a certain distance? If not, the spatial localization cannot help with computational efficiency.

2. The authors replied that it is hard to explain why 4DEnOI outperforms 4DVAR. But it would be nice to have some discussions or thoughts on this counterintuitive result. Is it possible that the static B at the initial time for 4DVAR is not as good as that in 4DEnOI, or the propagation of B during an assimilation window is harmful for some reasons?

Reviewer #4: Review of

Ensemble optimal interpolation for adjoint-free biogeochemical data assimilation

by Jann Paul Mattern and Christopher A. Edwards

General comments

The manuscript describes the application of an ensemble-like approach for coupled biogeochemical and physical data assimilation in marine environments. The ensemble used in the assimilation is composed of a static part (like in EnOI) with the addition of the assimilated trajectory-. If necessary, the static part of the ensemble can be regenerated. Results of the application of the proposed assimilation method (named 4dEnOI) for a relatively simple biogeochemical model have been compared with a previously existing 4dVar. Moreover, different settings of 4dEnOI are presented to investigate the effects of a number of elements of the assimilation method (e.g., localization, ensemble regeneration, inclusion of the assimilated trajectory in the ensemble, and iteration of the EnOI).

The manuscript is well written and changes in the revised manuscript try to respond to all the comments and suggestions raised in the previous review round. However, in my opinion the manuscript can be further improved considering its main strong and weak points.

The main strong points that I see are:

• The method proposed is a two-way strongly coupled biogeochemical-physical assimilation. Even if variable localization is applied, and thus the coupling between physical and biogeochemical variables is weakened, this is not a trivial task. In addition, the variable localization is one of the aspects investigated in the manuscript and the results show possible negative effects of strong coupling of all the physical and biogeochemical variables. In my opinion this aspect should be better highlighted in the Discussion & conclusions and in the abstract. Consider to clearly highlight also in the introduction that both physical and biogeochemical variables are assimilated in a coupled DA method.

• The manuscript proposes a method that has performances similar to the 4dVar and that does not require tangent linear or adjoint code. This point is already well declared in the Discussion & conclusions. This point could be further analyzed in the framework of operational applications. Is the assimilation method suitable to be implemented in an operational system? Is it more suitable than 4dVar?

On the other hand the revised manuscript is still affected by some weaknesses:

• The 4dEnOI is applied using a relatively simple biogeochemical model. Nothing against this choice but in the abstract and in the conclusions the results are presented citing largely more complex biogeochemical models (first two row of the abstract; and L. 541-543). I suggest to add some discussion about the limitations of using a “simple” NPDZ model and if there are elements that suggest that the results presented in the manuscript can be extended to complex biogeochemical models.

• The number of ensemble members is fixed to 25. Why this number of ensemble members? Do you expect that results will be different with a larger number of ensemble members? Maybe the fact of including the assimilated trajectory in the ensemble would be less relevant and the performances would be more similar to the ones of an EnOI, if a larger ensemble was adopted.

• The manuscript shows that the assimilation performances benefit from including the assimilated trajectory in the ensemble, however, similarly to the other Reviewers, I am not fully convinced about the mathematical consistency of the approach. Indeed, the ensemble that includes x1 has been not generated by a statistically consistent formulation. Maybe I am missing some points but I suggest the Authors to place 4dEnOI in the framework of the very comprehensive review by Carrassi et al., (2018) and to discuss possible limitations or drawbacks of the methods related to the fact that only one of the ensemble member is updated.

• After the first revision the manuscript has been modified to show that, even if the ensemble was not really static in the present applications, results would have been the same with a static ensemble (leaving apart x1). In my opinion, this point could be not fully understandable for the readers, therefore I suggest presenting the method as if the ensemble (from x2 to xn) is static in all the simulations except for the ones including regeneration. This approach requires changes in different parts of the manuscript but, in my opinion, it can considerably improve the presentation of 4dEnOI.

Specific comments

L. 44-53 The text from “(primarily construction” to “easy to implement” seems redundant and out of the main topic of the manuscript, since details are provided in Mattern et al. (2019). I suggest removing (or at least strongly reducing) this part.

L. 56-57 The EnKF is defined as the “best” among the ensemble-based DA techniques. Usually a DA approach can be better than others depending on the application, on the settings, on the model complexity. Thus, the claim that the EnKF is the “best” should be motivated considering the different available ensemble-based DA techniques and referring to relevant references.

L. 73 Change “(Verdy et al. 2016)” in “Verdy et al. (2016)”.

L. 74 “included in the computation” (“in” is missing).

L. 112 I suggest writing that observation locations are intended as spatial and temporal locations.

In the Fig. 2 caption, it should be added also regeneration: “the initial model state (ensemble generation) or a successive model state (ensemble regeneration) is used as the reference solution”.

L. 186 I suggest changing “localization” into “spatial localization”.

In my opinion section 3.1 should go in the Methods section. Moreover, observation errors are not presented in section 3.1 but it would be relevant to know how they are treated in 4dEnOI (and in 4dvar).

L. 264 Change “(Veneziani et al. 2009)” in “Veneziani et al. (2009)”.

In the last two rows of Table 1 chlorophyll glider observations are listed. Are these observations assimilated or used for validation?

A number of abbreviations are used in the headers of Table 2. I suggest explaing them in the caption.

In the Fig. 3 caption and at L. 309, I think that “normalized Jobs (eq. 7)” should be used instead of “cost function”.

L. 333 Commenting Fig. 4 the Authors refer to “chlorophyll-a” while in Fig. 4”phytoplankton is used.

Some patterns appear in the chlorophyll (phytoplankton increment) In Fig. 4. Are the satellite based chlorophyll observations affected by cloud coverage? If yes, are increments larger where observations are available? Cloud coverage and more generally some details about the temporal and spatial distribution of observations is missing (not only for chlorophyll).

L. 381-386 Fig. 5 shows that leaving out the x1 member the correlation of the ensemble increases more than leaving out any other ensemble member. Is this an intrinsic result of the fact that, being x1 corrected by DA, it is less correlated to the other static ensemble members? I suggest considering this aspect when commenting Fig. 5.

L. 388 I am not sure that “improved ensemble statistics” is correct. Indeed, it is not demonstrated that the mean or the covariance of the ensemble including x1 are better than those of the purely static ensemble. What the Authors can infer is based on normalized Jobs (ensemble performances).

Bibliography

Carrassi, A., Bocquet, M., Bertino, L., Evensen, G., 2018. Data assimilation in the geosciences: An overview of methods, issues, and perspectives. Wiley Interdiscip. Rev. Clim. Change 9, e535. https://doi.org/10.1002/wcc.535

7. PLOS authors have the option to publish the peer review history of their article (what does this mean?). If published, this will include your full peer review and any attached files.

Reviewer #1: No

Reviewer #4: No

---

## [Author Response · Author response to Decision Letter 1]

13 Jun 2023

The manuscript has undergone a second round of major changes based on the reviewer comments. A detailed description of the changes is contained in the accompanying response letter.

---

## [Decision Letter · Decision Letter 2]

22 Aug 2023

Ensemble optimal interpolation for adjoint-free biogeochemical data assimilation

PONE-D-22-03604R2

Dear Dr. Mattern,

We’re pleased to inform you that your manuscript has been judged scientifically suitable for publication and will be formally accepted for publication once it meets all outstanding technical requirements.

Kind regards,

Andrea Storto

Academic Editor

PLOS ONE

Additional Editor Comments (optional):

Reviewers' comments:

Reviewer's Responses to Questions

**Comments to the Author**

1. If the authors have adequately addressed your comments raised in a previous round of review and you feel that this manuscript is now acceptable for publication, you may indicate that here to bypass the “Comments to the Author” section, enter your conflict of interest statement in the “Confidential to Editor” section, and submit your "Accept" recommendation.

Reviewer #4: All comments have been addressed

2. Is the manuscript technically sound, and do the data support the conclusions?

Reviewer #4: Yes

3. Has the statistical analysis been performed appropriately and rigorously? 

Reviewer #4: Yes

4. Have the authors made all data underlying the findings in their manuscript fully available?

Reviewer #4: Yes

5. Is the manuscript presented in an intelligible fashion and written in standard English?

Reviewer #4: Yes

6. Review Comments to the Author

Reviewer #4: The Authors deeply revised the manuscript addressing all the comments. One minor point to integarte a new part of the manuscript:

L. 682 "As a next step, we are planning to apply the 4dEnOI technique to a biogeochemical model with more than 30 variables and a small ensemble size"

Is there any reference for this more complex biogeochemical model?

7. PLOS authors have the option to publish the peer review history of their article (what does this mean?). If published, this will include your full peer review and any attached files.

Reviewer #4: No

---

## [Editor Report · Acceptance letter]

25 Aug 2023

PONE-D-22-03604R2 

Ensemble optimal interpolation for adjoint-free biogeochemical data assimilation 

Dear Dr. Mattern:

I'm pleased to inform you that your manuscript has been deemed suitable for publication in PLOS ONE. Congratulations! Your manuscript is now with our production department. 

Kind regards, 

on behalf of

Dr. Andrea Storto 

Academic Editor

PLOS ONE